# Type B CTD Proteins Secreted by the Type IX Secretion System Associate with PorP-like Proteins for Cell Surface Anchorage

**DOI:** 10.3390/ijms23105681

**Published:** 2022-05-19

**Authors:** Dhana G. Gorasia, Christine A. Seers, Jacqueline E. Heath, Michelle D. Glew, Hamid Soleimaninejad, Catherine A. Butler, Mark J. McBride, Paul D. Veith, Eric C. Reynolds

**Affiliations:** 1Oral Health Cooperative Research Centre, Melbourne Dental School, Bio21 Institute, The University of Melbourne, Parkville, VIC 3010, Australia; gorasiad@unimelb.edu.au (D.G.G.); caseers@unimelb.edu.au (C.A.S.); jhea@unimelb.edu.au (J.E.H.); mglew@unimelb.edu.au (M.D.G.); cbutler@unimelb.edu.au (C.A.B.); 2Biological Optical Microscopy Platform (BOMP), The University of Melbourne, Parkville, VIC 3010, Australia; hamid.s@unimelb.edu.au; 3Department of Biological Sciences, University of Wisconsin-Milwaukee, Milwaukee, WI 53201, USA; mcbride@uwm.edu

**Keywords:** *Porphyromonas gingivalis*, periodontitis, *Flavobacterium johnsoniae*, type IX secretion system, type B cargo proteins/T9SS substrates/CTD proteins, cell-surface anchorage

## Abstract

The *Bacteroidetes* type IX secretion system (T9SS) consists of at least 20 components that translocate proteins with type A or type B C-terminal domain (CTD) signals across the outer membrane (OM). While type A CTD proteins are anchored to the cell surface via covalent linkage to the anionic lipopolysaccharide, it is still unclear how type B CTD proteins are anchored to the cell surface. Moreover, very little is known about the PorE and PorP components of the T9SS. In this study, for the first time, we identified a complex comprising the OM β-barrel protein PorP, the OM-associated periplasmic protein PorE and the type B CTD protein PG1035. Cross-linking studies supported direct interactions between PorE-PorP and PorP-PG1035. Furthermore, we show that the formation of the PorE-PorP-PG1035 complex was independent of PorU and PorV. Additionally, the *Flavobacterium johnsoniae* PorP-like protein, SprF, was found bound to the major gliding motility adhesin, SprB, which is also a type B CTD protein. Together, these results suggest that type B-CTD proteins may anchor to the cell surface by binding to their respective PorP-like proteins.

## 1. Introduction

Gram-negative bacteria have evolved a wide array of secretion systems to transport proteins to the cell surface or to the extracellular space. The type IX secretion system (T9SS) transports proteins across the outer membrane (OM) in members of the phylum *Bacteroidetes* [1,2,3,4,5,6]. This system was first identified in *Porphyromonas gingivalis*, a keystone pathogen of chronic periodontitis, and in *Flavobacterium johnsoniae*, a nonpathogenic soil bacterium. The *P. gingivalis* T9SS secretes cell surface virulence factors, such as gingipains [7], and the *F. johnsoniae* T9SS secretes cell-surface adhesins that are required for gliding motility, such as SprB, and many other proteins [7,8,9].

Proteins secreted by the T9SS have an N-terminal signal peptide that facilitates export across the inner membrane by the Sec system and have a conserved C-terminal domain, referred to as the CTD signal, that enables them to pass through the OM via the T9SS [10,11,12]. Two types of CTD signal, type A and type B have been defined and these differ in the T9SS components required for their export [13]. Most of the *P. gingivalis* secreted proteins have a type A CTD that target them to this system. Type B CTDs have recently been shown to target proteins, such as the major motility adhesin SprB, for secretion by the *F. johnsoniae* T9SS [14]. SprB is propelled rapidly along the cell surface by the gliding motility machinery driven by a rotary motor, resulting in the gliding movement of the cell [15,16]. Once on the surface, *P. gingivalis* type A CTD signals are removed by the sortase PorU and replaced with anionic-lipopolysaccharide (A-LPS), anchoring the proteins to the cell surface to form an electron dense surface layer (EDSL) [17,18]. At least some proteins with type B CTDs are also anchored to the cell surface, but the mechanism is not known.

The T9SS is composed of at least 20 proteins. In *P. gingivalis*, most are named with the “Por” prefix. Some of the orthologous *F. johnsoniae* T9SS proteins also use the Por prefix, whereas others use alternate names related to their roles in gliding motility. T9SS proteins (with the *F. johnsoniae* names given in brackets) include: PorK (GldK), PorL (GldL), PorM (GldM), PorN (GldN), Sov (SprA), PorT (SprT), PorU, PorW (SprE), PorP (multiple homologs including SprF), PorV, PorQ, PorZ, PorE, PorF, PorG, Plug/PG2092/PGN_0144, PorD/PGN_1783 and three transcription regulators PorX, PorY and SigP [7,19,20,21,22,23,24,25]. Recent studies have begun to reveal the structure and function of the T9SS components. The PorK-PorN-PorG complex forms a large ring-shaped structure [26]. The translocon complexes comprising SprA (an orthologue of *P. gingivalis* Sov), bound to either PorV or Plug, were solved using cryo-electron microscopy revealing the large 36-stranded SprA/Sov β-barrel to be the likely OM pore through which secreted proteins pass [24]. Recently, we showed that PorW bound to PorD forms a link between the Sov translocon and the PorK-PorN-PorG ring complex [25]. The PorL-PorM motor complex is anchored to the inner membrane [7,26,27] and has been shown to harvest the proton gradient and energise both secretion and motility [28]. The attachment complex comprising PorU-PorV-PorQ-PorZ anchors T9SS substrates to the cell surface via A-LPS using a sortase-like mechanism [29] [18,30]. PorE is an OM-associated lipoprotein localised in the periplasm [19]. Sequence analysis and computer modelling indicated PorE could be divided into four domains: a tetratricopeptide repeat (TPR) domain, a β-propeller domain, a carboxypeptidase regulatory domain-like fold and an OmpA C-terminus like putative peptidoglycan-binding domain. Recently, the crystal structure of the C-terminal domain of PorE bound to peptidoglycan was solved and PorE was suggested to play a role in anchoring the T9SS to the cell wall [31]. To achieve this, PorE would presumably bind to other components of the T9SS.

In this study, we characterised the binding partners of PorE using affinity purification mass spectrometry (AP-MS) to further understand PorE function in the T9SS. Our results show for the first time that PorE binds to the outer membrane β barrel protein PorP which, in turn, binds to the type B CTD protein, PG1035. We extended this study to *F. johnsoniae* and showed that the PorP-like protein, SprF binds to the type B CTD gliding motility protein, SprB. Collectively, these results suggest that type B CTD proteins are anchored to the cell surface bound to PorP-like proteins.

## 2. Results

### 2.1. PorP, PorE and PG1035 Form a Complex

To identify the potential binding partners of PorE, we expressed Strep-tagged PorE in the *P. gingivalis* W50 Δ*porE* mutant and called this strain PorE^Strep^. Immunoblot analysis of the PorE^Strep^ strain using streptactin II antibodies detected a major band at ~80 kDa, consistent with expression of the full-length fusion protein, with the expected molecular weights of the PorE and Strep tag being ~78 and ~1 kDa, respectively (Figure 1A). While T9SS mutants, including Δ*porE*, have beige pigmentation [19], the PorE^Strep^ strain was black pigmented on a blood agar plate, similar to WT (Figure 1B), indicating that the PorE-Strep fusion was functional. Strep-tag affinity capture was performed on cell lysates prepared from the PorE^Strep^ strain and the WT negative control. The captured proteins were quantitated by mass spectrometry and MaxQuant relative to the WT control. The label-free quantitation (LFQ) intensity ratios of PorE^Strep^:WT of the top 10 enriched proteins were plotted (Figure 1C). The LFQ intensity ratios of PorE, PorP and PG1035 were much higher than any other protein, suggesting these three proteins form a complex. The iBAQ metrics of MaxQuant can be used to estimate the relative abundance of different proteins in a sample. The iBAQ abundance of PorP and PG1035 were 1:1, whilst PorE was 10-fold more.

### 2.2. Chemical Cross-Linking Indicates Various Pairwise Interactions

To further understand the connection between these three proteins, we eluted the StrepTactin affinity captured material with desthiobiotin, cross-linked with disuccinimidyl sulfoxide (DSSO) and analysed the cross-linked proteins by Western blot. The immunoblot probed with anti-PorE and anti-PG1035 showed bands at ~82 and ~50 kDa, respectively, consistent with the expected PorE and PG1035 monomers as well as high molecular weight bands consistent with cross-linked products (Figure 2). The expected molecular weight of PorP was ~33 kDa. The bands at ~120 (anti-PorE only) and ~160 kDa (both antisera) may be consistent with PorE-PorP and PorE-PorP-PG1035 dimers and trimers, respectively (Figure 2). The band at ~80 kDa (anti-PG1035 only) may be consistent with PorP-PG1035, while the band at ~137 kDa (anti-PG1035) may be consistent with PorE-PG1035. The band at 137 kDa was not clearly evident using anti-PorE; however, considering the relative intensities of the ~137 and 160 kDa bands with anti-PG1035, a band corresponding to the putative PorE-PG1035 dimer would appear very faint in the anti-PorE blot. The band marked with an asterisk represents a degraded product of PG1035, as it was also present in the uncross-linked sample (Figure 2). Unfortunately, the PorP antibody we generated was nonreactive and could not be used in this analysis. Together, bands could tentatively be assigned to all possible heterodimers and to the complete trimeric complex.

### 2.3. PG1035 Complex Formation Is Independent of PorU and PorV

Next, we explored whether the formation of this complex is reliant on other components of the T9SS. Since PG1035 is a type B CTD protein, we analysed the Δ*porU* and Δ*porV* deletion mutant strains, because it was previously shown that secretion of *F. johnsoniae* type B CTD proteins does not require PorU and PorV [32]. We performed BN-PAGE immunoblot analysis using PorE antibodies on the above *P. gingivalis* strains together with the controls of WT, ABK^−^ (strain lacking all three gingipains) and a Δ*pg1035* deletion mutant (see below). In the WT, ABK^−^ and Δ*porU* strains a band at ~420 kDa was observed, which likely corresponded to the PorE-PorP-PG1035 complex (plus detergent), whilst in the Δ*pg1035* mutant, the band at ~420 kDa was missing and a band at ~260 kDa was observed instead, which may represent the PorE-PorP subcomplex (Figure 3A). In the Δ*porV* mutant, two bands were observed, one at ~420 kDa and the other at ~260 kDa, suggesting that the PorE-PorP-PG1035 complex can also form in the Δ*porV* mutant. Since two bands were detected in the Δ*porV* mutant and the ~260 kDa band was more intense, this suggests that the formation of the PorE-PorP-PG1035 is less efficient in the absence of PorV. This finding was further validated using a Δ*porV* mutant in the 381 *P. gingivalis* strain. The choice of this strain is explained below. In this strain, the band at ~420 kDa was the most prominent, confirming that PorV is not required for the formation of the PorE-PorP-PG1035 complex (Figure 3A). Together, these results suggest that PorE-PorP-PG1035 complex formation is independent of PorU and PorV.

We endeavoured also to test complex formation in the Δ*porP* mutant. Screening of this mutant by denaturing immunoblot analysis using antibodies for PorE failed to produce a band at its expected MW of ~78 kDa, and instead produced a band at ~60 kDa (Figure 3B). For this analysis, the ABK^−^ strain was selected as the control to fully ensure that PorE was not degraded by the gingipains. PorE was detected at its expected MW of ~ 78 kDa in the ABK^−^ strain. This suggests that in the absence of PorP, PorE is unstable or vulnerable to proteolysis, hence, preventing the use of the Δ*porP* mutant for studies requiring functional or intact PorE.

### 2.4. The PorP-PG1035 Complex Can Form Independently of PorE

From the BN-PAGE immunoblots above, a putative PorE-PorP complex formed in the absence of PG1035 (Figure 3A). Next, we tested whether a PorP-PG0135 complex can form in the absence of PorE. To answer this, we performed co-immunoprecipitation on WT, Δ*porE* and Δ*pg1035* mutants using PG1035 antibodies. The immunoprecipitated material was quantified by mass spectrometry and MaxQuant relative to the Δ*pg1035* negative control (Table 1). As expected, the LFQ intensity ratios of PG1035, PorP and PorE were highest in the WT (Table 1), Appendix A. In the Δ*porE* mutant strain, PorP was found enriched together with PG1035 in the pull down (Table 1), confirming a direct interaction between PorP and PG1035 and suggesting that the PorP-PG1035 complex can form independently of PorE. Furthermore, small amounts of PorK and PorN were also found in the immunoprecipitated material in both the WT and Δ*porE* mutant strains suggesting that the PorE-PorP-PG1035 complex may be associated with the PorK/N rings.

### 2.5. PG1035 Is a Non-Essential Component of the T9SS

Since PG1035 complexed with the PorE and PorP components of the T9SS, we examined whether PG1035 is required for T9SS function. We created a Δ*pg1035* deletion mutant in *P. gingivalis* W50. This mutant was black pigmented (Figure 4A) and an immunoblot with A-LPS antibodies showed high molecular weight A-LPS similar to WT (Figure 4B), indicating that the CTD proteins were modified with A-LPS. Most mutants that are defective for T9SS function are apigmented, and they fail to attach A-LPS to proteins. Thus, A-LPS blot bands above 60 kDa are usually not detected. The results suggest that although PG1035 bound to PorP and PorE, it was not essential for T9SS function under the conditions tested. Moreover, we found that expression of PG1035 was strain dependent (Figure 4C). It was expressed in *P. gingivalis* strains W50 and 381 but not in strain ATCC 33277 (Figure 4C) [33]. This further supports the nonessential role of PG1035 in the T9SS, since the ATCC 33277 strain has a functional, well-characterised T9SS [7].

### 2.6. PG1035 Is Localised on the Cell Surface and Is Not Modified by A-LPS

Type A CTD proteins in *P. gingivalis* covalently bind to A-LPS and are anchored onto the cell surface. To determine if the type B CTD protein, PG1035, is localised on the cell surface, we performed immunofluorescence microscopy. Antisera against PG1035 were used to detect PG1035 on WT cells and on cells of Δ*pg1035* mutants. The fluorescent signal was observed at high intensity on the WT cell surface compared to the Δ*pg1035* mutant negative control (Figure 5A,B). Quantification of the fluorescent signal using ImageJ revealed considerably higher intensity of PG1035 antibody signal on WT compared to Δ*pg1035* mutant cells (Figure 5C). This suggests that PG1035 is located on the cell surface. To examine if PG1035 is modified with A-LPS, an immunoblot on the cell lysates of the WT and of the Δ*pg1035* mutant strain was performed using PG1035 antibodies. PG1035 was only detected in the WT and only at the molecular weight of ~50 kDa (Figure 5D), suggesting that PG1035 was not modified with A-LPS and that its CTD was not cleaved. In contrast, A-LPS-modified type A CTD proteins are often detected as ladders starting ~20 kDa above the expected molecular weight of the unmodified protein [7,17]. Collectively, these results suggest that PG1035 is anchored onto the cell surface by binding to the outer membrane β-barrel protein, PorP, rather than by covalent linkage to A-LPS.

### 2.7. The F. johnsoniae Motility Protein SprB Binds to the PorP Homologue SprF

PG1035 is the only *P. gingivalis* protein that has a type B CTD [14]. In contrast, *F. johnsoniae* has twelve type B CTD proteins, including the major gliding motility protein SprB. SprB is co-expressed with the PorP orthologue, SprF, which is essential for the secretion and cell-surface localisation of SprB [14]. What SprB interacts with on the cell surface and how SprB interacts with the motility machinery are not known. Since the *P. gingivalis* type B CTD protein PG1035 is likely bound to PorP on the cell-surface, we reasoned that SprB may be bound to the PorP orthologue, SprF, on the *F. johnsoniae* cell surface. To test this hypothesis, we performed co-immunoprecipitation on a *F. johnsoniae* strain co-expressing SprF and the CTD of SprB fused to sfGFP (sfGFP-CTD_SprB (368 aa)_) [14]. The sfGFP in this strain was previously shown to be rapidly propelled along the cell surface demonstrating its successful secretion by the T9SS and association with the motility machinery [14]. The material immunoprecipitated by anti-GFP antibodies was quantitated by mass spectrometry and MaxQuant relative to a *F. johnsoniae* WT control. The LFQ intensity ratios and iBAQ abundances of sfGFP-CTD_SprB (368 aa)_ relative to the *F. johnsoniae* WT were plotted for the top 30 proteins (Figure 6A,B). The highest LFQ intensity ratios were of SprF and sfGFP-CTD_SprB (368 aa)_, suggesting that SprF and SprB interact directly through the C-terminal 368 aa of SprB. The iBAQ abundances were similar, suggesting a 1:1 stoichiometry between these two proteins. Other known components of the gliding/T9SS machinery such, as SprA, GldJ, Fjoh_4997 (PPI), SprE, GldN and GldK as well as the putative new component, Fjoh_3466 (homolog of PorD), which interacts with PorW [25], were also specifically pulled down but at much lower abundances (Figure 6A,B). Figure 6C summarises the potential arrangement of the SprB-associated proteins and is discussed further below. Together, these results suggest that SprB may be held on the surface and brought into contact with the rest of the motility machinery via a direct interaction with the PorP ortholog, SprF.

## 3. Discussion

The *Bacteroidetes* T9SS transports many proteins to the cell surface or releases them in the culture fluid. These proteins harbour either a type A or type B CTD signal at their C-terminus. Type A CTD proteins attach to the cell surface via covalent linkage to A-LPS [17,18,30]; however, very little was known regarding attachment of type B CTD proteins to the cell surface. Herein we provide evidence suggesting that type B CTD proteins attach to the cell surface by interacting with PorP-like proteins.

We identified a novel T9SS complex consisting of PorP, PorE and the type B CTD protein PG1035, and we obtained evidence for a stable complex between PorE and PorP that was independent of PG1035 (Figure 2 and Figure 3). We also provide evidence for a direct interaction between PorP and PG1035 that was independent of PorE (Figure 2, Table 1). Therefore, the architecture of the mature complex is likely to be PorE bound to PorP bound to PG1035 (Figure 7A). PorP may bind to one of the two most N-terminal domains of PorE, which are a TPR domain and a β-propeller domain [19]. TPR and β-propeller domains in other proteins frequently function as structural scaffolds involving protein–protein interactions [34,35]. Since the C-terminal domain of PorE binds to peptidoglycan [31], PorE likely forms a bridge between the peptidoglycan cell wall and the OM β-barrel protein PorP (Figure 7A) [7,27,36].

PG1035 is the only *P. gingivalis* protein with a type B CTD [14]. Given that all characterised proteins containing either type A or type B CTDs are secreted and anchored onto the cell surface or released in the culture fluid, it is likely that PG1035 is translocated across the OM and then attached to the cell surface via PorP. We identified a direct interaction between PorP and PG1035 and found that PG1035 is localised to the cell surface. The anchorage to PorP is likely to be through the CTD of PG1035, as its CTD was observed to be intact in the pull-down, and only the CTD is known to be conserved amongst type B CTD proteins. Surprisingly, in the absence of PorE, the PorP-PG1035 complex could still form suggesting that PorE may not be required for the secretion of PG1035. Cross-linking suggested that PorE may also directly interact with PG1035, which could either occur through the lumen of the PorP barrel, or else be a non-essential interaction prior to PG1035 secretion. PG1035 was not essential for T9SS function under laboratory conditions and we therefore speculate that it may have a stabilising or protective function as a cap for PorP.

PorK and PorN were also found associated with the PorE-PorP-PG1035 complex in small amounts. Previously, Vincent et al. showed that PorP interacts with PorK using co-immunoprecipitations in *Escherichia*
*coli* [27]. Our work supports these findings and suggests a role for the PorP-PorE-PG1035 complex in anchoring the PorK/N rings to the peptidoglycan cell wall, as PorE is shown to bind to the peptidoglycan [31] (Figure 7A).

The CTD of PG1035 is similar to the CTD of *F. johnsoniae* SprB, and those of 11 other *F. johnsoniae* type B CTD proteins. In contrast to *P. gingivalis*, which encodes a single PorP, *F. johnsoniae* encodes 10 PorP-like proteins, and these are thought to be specific for secretion of their cognate type B CTD proteins [14,32]. SprB, for example, requires its cognate PorP-like protein, SprF, for secretion and attachment on the cell surface. SprB is a highly repetitive 669 kDa protein that moves rapidly along the cell surface, resulting in gliding motility. How this protein interacts with the secretion and motility machine(s) is not known. We have shown that sfGFP-CTD_SprB(368 aa)_ and SprF form a complex, implicating the C-terminal 368 aa of SprB in this interaction. The interaction could be either at the periplasmic side of the OM or on the cell surface. Given that GFP was observed on the cell surface in this strain [14], it is likely the interaction occurs there. Together, this suggests that SprB is anchored to the cell surface through SprF. Extending this, it would appear that one function of PorP-like proteins is to anchor their respective type B CTD proteins to the cell surface. In *F. johnsoniae*, six of the *porP*-like genes are located next to *porE*-like genes [14]. Notably, *sprB* and the adjacent *sprF* (*porP*-like) are not located next to a *porE*-like gene. We postulate that mobile type B CTD proteins, such as SprB, are anchored only to PorP-like proteins (Figure 7B), whereas nonmobile type B CTD proteins are further anchored to the cell wall via PorP-PorE-like complexes.

The motility protein, GldJ, was also specifically pulled down with SprB but at a lower abundance than SprF. GldJ is an OM lipoprotein paralogous to GldK that is required for propulsion of SprB on the cell surface. It appears to localise in a helical manner, and it was proposed to be associated with the track on which SprB is propelled [37,38]. SprF may link SprB, directly or indirectly, to this track, which we recently proposed to also include the GldK and GldN proteins [39] (Figure 7B) that were pulled down at a slightly higher abundance than GldJ (Figure 6).

SprB was also found associated with SprA, PPI, Fjoh_3466 (PorD) and SprE (Figure 6). These interactions may have been captured during the translocation of SprB through the SprA channel. Previously, SprA was shown to interact with the peptidyl-prolyl *cis-trans* isomerase (PPI) [24] which is consistent with our findings. SprA is known to associate with either PorV or the Plug [24], but in the SprB pull-down neither of these proteins were observed (Figure 6C). This is not surprising since SprB secretion is dependent on SprA but independent of PorV and, instead, requires SprF [8,32]. Similarly, *P. gingivalis* PorE-PorP-PG1035 complex formation was also independent of PorV (Figure 3). This suggests that PG1035, like other type B CTD proteins, is secreted independently of PorV.

PorV has been proposed to collect the type A CTD proteins from the Sov translocon and shuttle them to the attachment complex for covalent binding to A-LPS and cell surface anchorage [24,29]. PorV and the PorP-like proteins are all 14-stranded OM β-barrels of the FadL family that independently bind to type A or B CTD proteins, respectively. We postulate that PorV and PorP-like proteins (SprF) bind to the same site on SprA/Sov to collect their respective cargo proteins. When SprB or PG1035 is being secreted, SprF or PorP, rather than PorV, would be complexed with SprA/Sov to collect its respective substrate(s) (Figure 7).

In conclusion, we showed, for the first time, that the lipoprotein PorE associates with the outer membrane β-barrel protein, PorP, which in turn interacts with a type B CTD protein, PG1035 (Figure 7A). In *F. johnsoniae*, we demonstrated that the type B CTD protein, SprB, is anchored to the cell surface by interacting with the PorP-like protein, SprF (Figure 7B). Collectively, these findings suggest that PorP-like proteins anchor type B CTD proteins to the cell surface. Further work is required to determine if PorP-like proteins collect their respective type B substrates directly from the Sov translocon.

## 4. Materials and Methods

### 4.1. Bacterial Strains and Culture Conditions

*P. gingivalis* was grown in tryptic soy-enriched brain heart infusion broth (TSBHI) (25 g/L Tryptic soy, 30 g/L BHI) supplemented with 0.5 mg/mL cysteine, 5 μg/mL haemin and 5 μg/mL menadione. For blood agar plates, 5% defibrinated horse blood (Equicell, Bayles, Australia) was added to enriched trypticase soy agar. Mutant strains (i.e., Δ*porU*, Δ*porV* and Δ*porP*) were grown in the same media as above with the appropriate antibiotic selections [7,17,40]. All *P*. *gingivalis* strains were grown anaerobically (80% N_2_, 10% H_2_ and 10% CO_2_) at 37 °C. *F. johnsoniae* UW101 was grown in Casitone Yeast Extract (CYE) medium [41] at 25 °C with shaking. Twenty micrograms per milliliter tetracycline was included in the growth medium when required.

### 4.2. Generation of P. gingivalis pg1035 Deletion Mutant

A *P. gingivalis* W50 Δ*pg1035* mutant was produced by replacing *pg1035* codons (coding from Met to stop) with an *ermF* gene under the control of its own promoter. To do this, the suicide plasmid pPG1035ermF was produced as follows. Two PCR amplicons were generated using *P. gingivalis* W50 DNA as a template. PCR (1) used oligonucleotides pg1035fl_for (GGTTTGGATCCGAAGACTTC) and pg1035-ermF_rev (AAGCAATAGCGGAAGCTATCCCGATTGTTTTTCTTTTTTATCC) giving a 351 bp product. PCR (2) used pg1035fl_rev (GCGCTCCATACCACCACCGA) and ermF-pg1035_for (**GAAAAATTTCATCCTTCGTAG**CAAAAAACGAATCGACCTTTC) giving a 346 bp product. A third PCR (3) used the *ermF* gene in pAL30 as a template with oligonucleotides ermFprom_for (GATAGCTTCCGCTATTGCTT) and ermFstop_rev (**CTACGAAGGATGAAATTTTTC**AGG). PCR (1) was annealed to PCR (3) by the complementary nucleotides (underlined in pg1035-ermF_rev and ermFprom_for) and spliced using pg1035fl_for and ermFstop_rev external primers producing Splice1. Splice1 was annealed to PCR (2) via complementary nucleotides (bold in ermF-pg1035_for and ermFstop_rev) and spliced using external primers pg1035fl_for and pg1035fl_rev, giving a final 1740 bp Splice2 product. Splice2 was cloned into pGem-T Easy (Promega) and integrity of the DNA confirmed by sequencing. The plasmid was linearised using SacI and transformed into electrocompetent *P. gingivalis* W50. Erythromycin-resistant *P. gingivalis* recombinants were selected from agar plates supplemented with 10 µg/mL erythromycin and correct genome insertion of *ermF* at the *pg1035* locus was assessed using PCR with primers pg1035fl_for2 (agtatctcttttgtgacgag) and pg1035fl_rev2 (actcagcccttttatctcgt) that anneal external to the recombination cassette integration site.

### 4.3. Generation of Strep-Tag II PorE

The PorE fusion protein with a C-terminal Strep-tag™ II WSHPQFEK (PG1058StrepII) was produced for use in immunoprecipitation. To do this, the suicide plasmid p1058cepA [19] was used as a template for PCR, with primers PG1058DomIIIFor1 (CAGACCCGGGAATATGGGACAACCGGTC) and PG1058StrepIIRev (TTCCCGCGGGAGCGATTA**TTTTTCGAACTGCGGGTGGCTCCA**ACGCAACTCTTCTTC) producing an amplicon that included a 3′ portion of the pg1058 ORF with a 5′ XcmI site (underlined), and at the 3′ end, the Strep-tag™ II codons (bold), a stop codon and then a SacII site (underlined). The amplicon was digested with XcmI and SacII and ligated to XcmI/SacII-digested p1058cepA to generate p1058strepIIcepA. Plasmid p1058strepIIcepA was digested with DraIII and used to transform the *P. gingivalis* W50 Δ*porE* mutant [19] with recombination at the *mfa1* locus. The *1058strepII* was under the control of the *mfa1* promoter. Transformants were selected on HBA containing erythromycin (10 µg/mL) and ampicillin (5 µg/mL). Correct genomic recombination was confirmed using PCR and one clone was designated as PorE^Strep^.

### 4.4. PG1035 Antibody

A rabbit polyclonal antibody against PG1035 (66–496 aa) was generated by Genscript according to their SC1676 package.

### 4.5. Blue Native Gel Electrophoresis

BN-PAGE was performed essentially as described [7]. Briefly, *P. gingivalis* cells were pelleted by centrifugation at 5000× *g* for 5 min at 4 °C and the pellet was suspended in native gel buffer containing 1% n-dodecyl-β-D-maltoside (DDM), complete protease inhibitors (Roche) and 5 mM MgCl_2_. After sonication (3 × 15 s) [7], the samples were clarified by centrifugation at 16,000× *g* for 20 min at 4 °C. Coomassie Blue G-250 was added to the samples at a final concentration of 0.25% and the samples were electrophoresed on nondenaturing Native PAGE Novex 3–12% Bis-Tris Gels. The proteins in the gels were transferred onto a PVDF membrane for immunodetection with antibodies specific for PorE as per the immunoblot method below [7].

### 4.6. Immunoblot Analysis

Cell lysates from *P. gingivalis* were separated by SDS-PAGE and the proteins were transferred onto a nitrocellulose membrane. The membrane was blocked in 5% skim milk for 1 h and probed with, mAb-1B5 (1:200) (a kind gift from M. A. Curtis) [42], PorE [19] (1:3000), PG1035 (1:1000) or Strep-tag II (1:2000) specific antibodies for 1 h at room temperature followed by anti-mouse or anti-rabbit HRP-conjugated secondary antibodies (1:3000) for 1 h. The signal was developed using SuperSignal West Pico chemiluminescent substrate and visualised with a LAS-3000 imaging system.

### 4.7. Co-Immunoprecipitation

*P. gingivalis* cells were lysed in 20 mM Tris-HCl, pH 8, 100 mM NaCl, 1% DDM, tosyl-L-lysyl-chloromethane hydrochloride (TLCK) and complete protease inhibitors. The cells were then sonicated [7] and clarified by centrifugation at 16,000× *g* for 10 min at 4 °C. Cell lysates were mixed with either StrepTactin Sepharose (GE Healthcare) for PorE^Strep^ immunoprecipitation, PG1035 antibodies bound to agarose for PG1035 co-immunoprecipitation or anti-GFP agarose (Chromotek) for sfGFP_SprB (368 aa)_ immunoprecipitation. The beads were rotated on a wheel for 2 h at 4 °C then washed with 20 mM Tris-HCl, pH 8, 100 mM NaCl, followed by 300 mM salt washes and lastly with 10 mM Tris-HCl, pH 8. The beads were suspended in Laemmli buffer and boiled for 10 min. The samples were subjected to SDS-PAGE electrophoresis for 6 min and stained with Coomassie Blue and subjected to in-gel trypsin digestion [43]. Tryptic peptides were extracted from the gel pieces using 50% acetonitrile in 0.1% TFA and sonicated for 10 min in a sonicator bath. The samples were concentrated in a vacuum centrifuge before analysis using LC-MS/MS.

### 4.8. LC-MS/MS

The tryptic peptides were analysed by LC-MS/MS using the Q Exactive Plus Orbitrap mass spectrometer coupled to an ultimate 3000 UHPLC system (Thermo Fisher Scientific). Buffer A was 2% acetonitrile and 0.1% formic acid, and buffer B consisted of 0.1% formic acid in acetonitrile. Sample volumes of 1 µL were loaded onto a PepMap C18 trap column (75 µM ID × 2 cm, 100 Å) and desalted at a flow rate of 2 µL/min for 15 min using buffer A. The samples were then separated through a PepMap C18 analytical column (75 µM ID × 15 cm, 100 Å) at a flow rate of 300 nL/min, with the percentage of solvent B in the mobile phase changing from 2 to 10% in 1 min, from 10 to 35% in 50 min, from 35 to 60% in 1 min and from 60 to 90% in 1 min. The spray voltage was set at 1.8s kV, and the temperature of the ion transfer tube was 250 °C. The S-lens was set at 50%. The full MS scans were acquired over a m/z range of 300–2000, with a resolving power of 70,000, an automatic gain control (AGC) target value of 3 × 10^6^, and a maximum injection of 30 ms. Dynamic exclusion was set at 90 s. Higher energy collisional dissociation MS/MS scans were acquired at a resolving power of 17,500, AGC target value of 5 × 10^4^, maximum injection time of 120 ms, isolation window of *m*/*z* 1.4, and NCE of 25% for the top 15 most abundant ions in the MS spectra. All spectra were recorded in profile mode.

The relative abundances of proteins were quantified by MaxQuant (Ver.1.5.3.30) [44]. Raw MS/MS files were searched against the *P. gingivalis* W50 or the *F. johnsoniae* UW101 database with the sequence of sfGFP added. The default MaxQuant parameters were used except LFQ min ratio count was set to 1 and the match between runs was selected. MaxQuant normalised the data set as part of data processing. For quantification of the immunoprecipitated material, the LFQ intensity ratio test strain/control strain was used to identify proteins that were significantly enriched. To determine the relative abundance of the enriched proteins, the corrected iBAQ intensity (test strain–control strain) was calculated and expressed as a percentage relative to the target protein.

### 4.9. Cross-Linking

Cell lysates from PorE^Strep^ were mixed with StrepTactin Sepharose as above. The beads were washed in PBS and StrepTactin Sepharose bound material was eluted using 5 mM desthiobiotin. Disuccinimidyl sulfoxide (DSSO) cross-linker was added at a final concentration of 1 mM and the reaction was incubated for 15 min at room temperature. Twenty mM of Tris, pH 8.0 was added to stop the reaction. The cross-linked material was analysed by immunoblot using PorE and PG1035 specific antibodies as described above. A lower amount of protein was loaded for the uncross-linked sample to ensure sharper bands.

### 4.10. Fluorescence Microscopy

Fluorescence microscopy was essentially performed as described in Glew et al. [45]. Briefly, *P. gingivalis* WT and the Δ*pg1035* mutant cells were washed in 140 mM NaCl, 12 mM K_2_HPO_4_, 14 mM Na_2_HPO_4_, 25 mM glucose (pH 7.5) (Pg-PBS buffer) and resuspended to the same optical density at 650 nm (OD_650_). For fluorescent staining of PG1035, cells were blocked with 3% BSA in Pg-PBS containing 10 mM TLCK for 1 h at 4 °C, incubated with anti-PG1035 (1:200) for 40 min at 4 °C, washed twice in Pg-PBS, incubated with Alexa Fluor 647-conjugated goat anti-rabbit mouse IgG (1:200) for 40 min at 4 °C (Thermo Fisher Scientific) and washed again. Immunostained cells were incubated with Syto9 and adhered to poly-l-lysine-treated coverslips. Images were captured on a DeltaVision Elite microscope (Olympus U-Plan S-Apo 100×/1.42 NA objective; excitation at 475 nm and emission at 523 nm for Syto9; excitation at 632 nm and emission at 676 nm for Alexa Fluor 647), deconvolved using softWoRx 6.1 software, and analysed using ImageJ software. Quantitative fluorescence intensity analysis was performed using the total raw intensity of the Alexa-647 (red) and Syto9 (green) stains from four random fields of view of PG1035 fluorescently stained WT and *pg1035* strains using ImageJ software. Their corresponding intensity ratios (total raw intensity of the red channel divided by the total raw intensity of the green channel) were graphed using box plot.

## Figures and Tables

**Figure 1 ijms-23-05681-f001:**
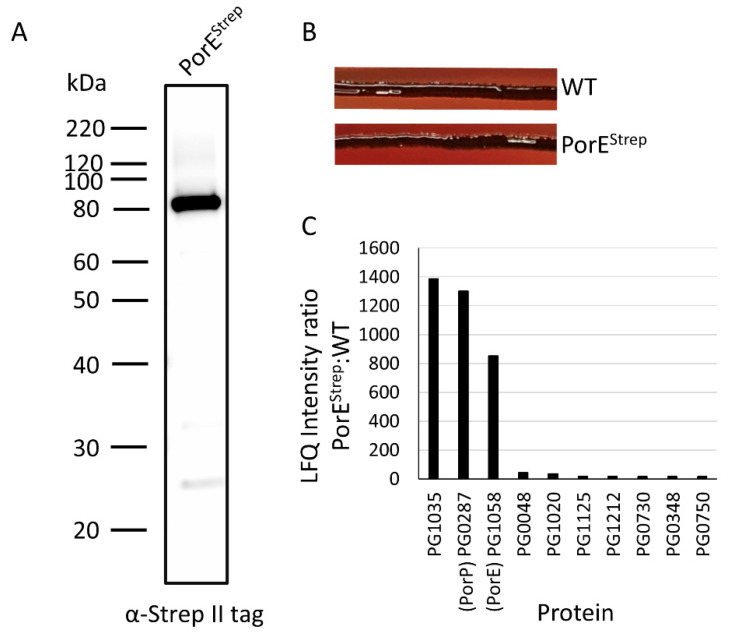
PorP, PorE and PG1035 form a complex: (**A**) cell lysates from the *P. gingivalis* PorE^Strep^ strain were subjected to immunoblot analysis using antibodies against the strep II tag; (**B**) black pigmentation of *P. gingivalis* WT and PorE^Strep^ strains on a blood agar plate after 7 days; (**C**) *P. gingivalis* WT and PorE^Strep^ tagged strains were subjected to affinity capture using StrepTactin Sepharose beads. The affinity-captured proteins were digested with trypsin and analysed by mass spectrometry and quantified using MaxQuant software. The ratio of LFQ intensities of PorE^Strep^ to WT was plotted.

**Figure 2 ijms-23-05681-f002:**
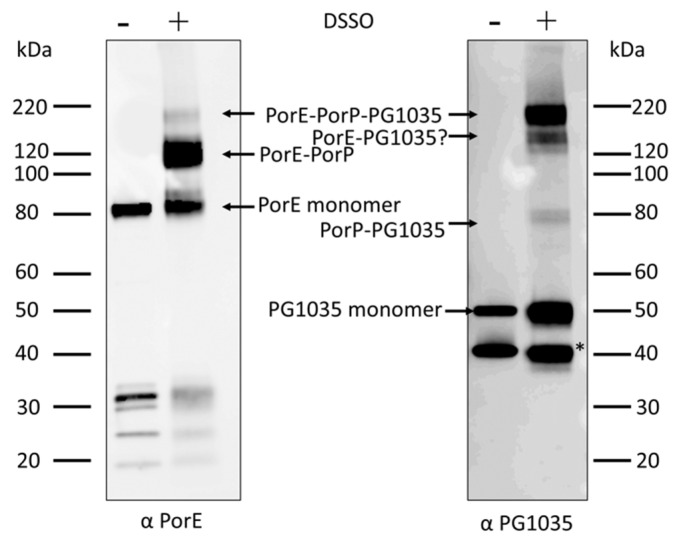
Direct interactions between PorE-PorP and PorP-PG1035. The immunoprecipitated material from StrepTactin Sepharose beads from *P. gingivalis* PorE^Strep^ strain was eluted with desthiobiotin and cross-linked with DSSO. Immunoblot analysis was performed on the cross-linked samples using antibodies specific to PorE and PG1035. The asterisk shows degraded product of PG1035.

**Figure 3 ijms-23-05681-f003:**
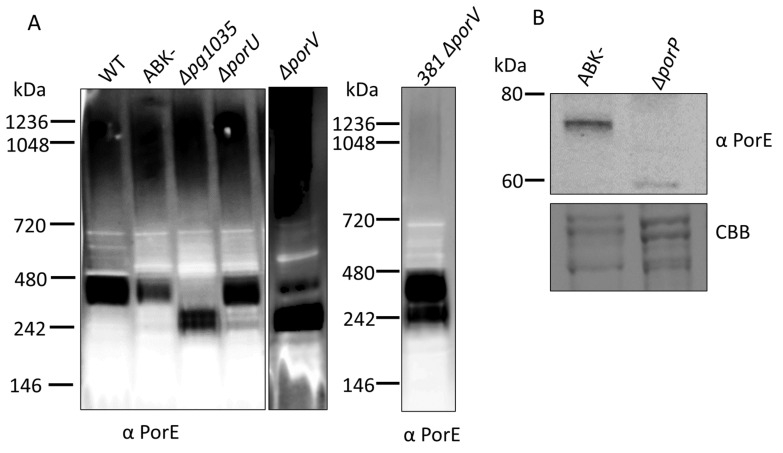
Native migration of the PorP-PorE-PG1035 complex and stabilisation of PorE by PorP: (**A**) whole cell lysates from WT, ABK^−^, Δ*pg1035*, Δ*porU* and Δ*porV* mutants and 381 Δ*porV* were electrophoresed on a BN-PAGE gel, and the proteins were transferred onto a PVDF membrane and probed with anti-PorE antibodies; (**B**) whole cell lysates from *P. gingivalis* ABK^−^ and Δ*porP* mutant were electrophoresed on an SDS-PAGE gel. The proteins were transferred onto a nitrocellulose membrane and probed with anti-PorE antibodies. Coomassie blue (CBB) stained gel shows the relative loading amount.

**Figure 4 ijms-23-05681-f004:**
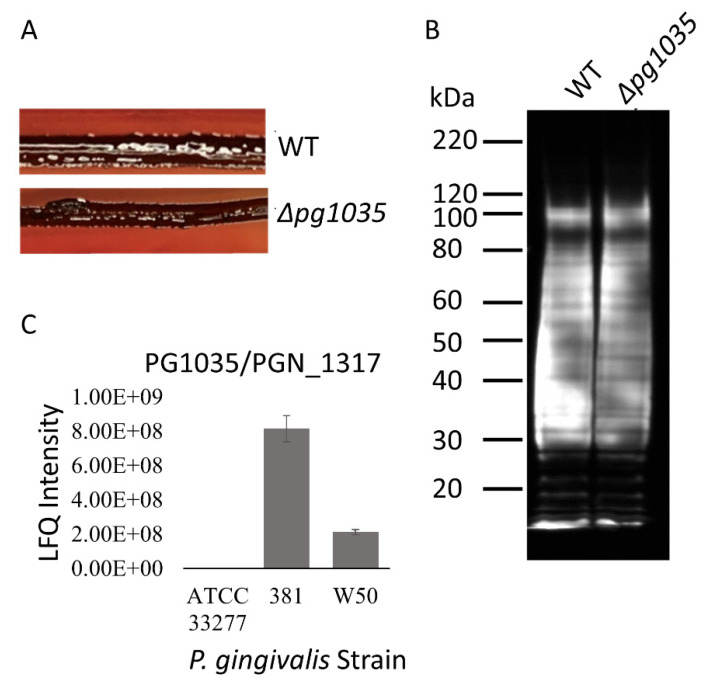
PG1035 is a nonessential component of the T9SS. (**A**) Black pigmentation of the WT and Δ*pg1035* deletion mutant on blood agar plate after 7 days. (**B**) Proteins in the whole cell lysates of the WT and Δ*pg1035* deletion mutant were electrophoresed on an SDS-PAGE gel. The proteins were transferred onto a membrane and probed with mAb-1B5 (anti-A-LPS) antibodies. (**C**) Expression levels of PG1035 (ATCC 33277 PGN_1317) in different *P. gingivalis* strains measured by mass spectrometry and quantitated by MaxQuant.

**Figure 5 ijms-23-05681-f005:**
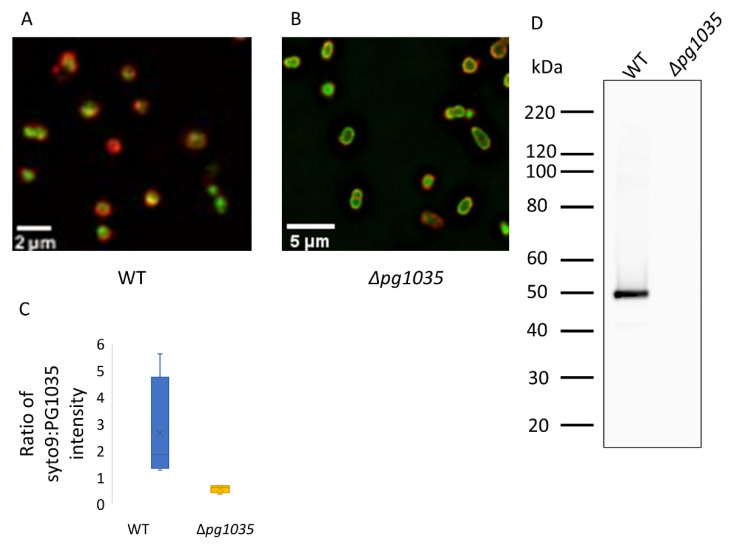
PG1035 is localised on the cell surface and is not modified by A-LPS. (**A**) WT and (**B**) Δ*pg1035* mutant cells were immune stained with anti-PG1035 antibodies, followed by goat anti-rabbit secondary antibodies conjugated to Alexa-647 (red) and syto 9 (green). (**C**) Box plot of the fluorescent intensity signal measured by ImageJ. (**D**) Proteins in the whole cell lysates of the WT and Δ*pg1035* deletion mutant were electrophoresed on an SDS-PAGE gel. The proteins were transferred onto a membrane and probed with PG1035 antibodies.

**Figure 6 ijms-23-05681-f006:**
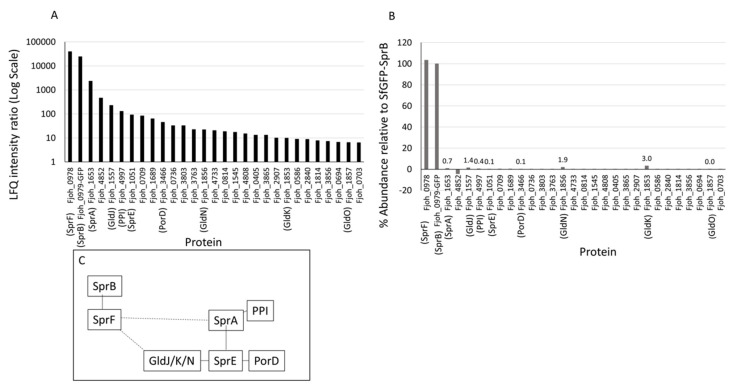
The *F. johnsoniae* type B CTD protein SprB and the PorP-like protein SprF form a complex. *F. johnsoniae* WT and sfGFP-CTD_SprB (368 aa)_ strains were subjected to co-immunoprecipitation with anti-GFP agarose beads. The immunoprecipitated samples were digested with trypsin and analysed by mass spectrometry and quantified using MaxQuant software. (**A**) The ratio of LFQ intensities of sfGFP-CTD_SprB (368 aa)_ to WT and (**B**) % abundance based on iBAQ intensities were plotted. (**C**) Summary of the protein interactions. Solid lines: observed interactions with high confidence including those found in the homologous proteins in *P. gingivalis* [24,25]; dashed lines: predicted interactions.

**Figure 7 ijms-23-05681-f007:**
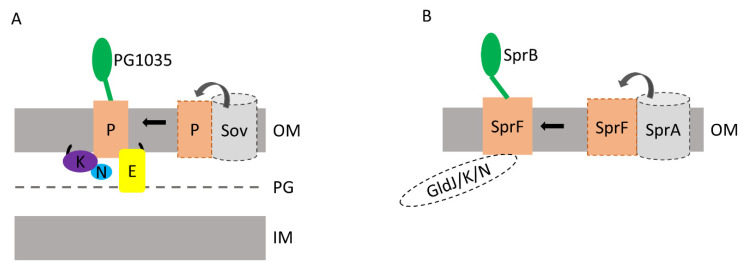
The proposed organisation of PorE-PorP-PG1035 and SprF-SprB complexes. (**A**) PorE (E) is localised in the periplasm tethered to the OM via its lipid moieties (lipid shown as black line). PorE interacts with the OM 14-stranded β-barrel protein, PorP (P). PorP associates with PG1035 on the cell surface via the CTD of PG1035, shown as a green line. PorP also associates with the PorK/N (K/N) ring via its interaction with PorK (K) lipoprotein. (**B**) The gliding motility protein, SprB, associates with the OM 14-stranded β-barrel protein, SprF, on the cell surface. SprF may also associate with the Gld/J/K/N protein track. Based on our results, we predict that PorP and PorP-like proteins such as SprF associate with the Sov/SprA translocon to collect their respective type B CTD proteins. OM: outer membrane; PG: peptidoglycan; IM: inner membrane.

**Table 1 ijms-23-05681-t001:** PG1035 complexes in WT and *porE* mutant.

Protein	WT		*porE*
	LFQ	iBAQ (%)	iBAQ (%)
PG1035	7266	100	100
PorE	589	17	0
PorP	1082	31	14
PorK	3	0.1	0.1
PorN	3	0.2	0.1

## Data Availability

Data are available upon request from the corresponding authors.

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
