# Peer review of "Type B CTD Proteins Secreted by the Type IX Secretion System Associate with PorP-like Proteins for Cell Surface Anchorage"

_ijms, 2022, doi:10.3390/ijms23105681_

Round 1

Reviewer 1 Report

The present article :Type B CTD proteins secreted by the type IX secretion system associate with PorP-like proteins for cell surface anchorage"  highlighted PorE and PorP components of the T9SS secretory system and  how type B CTD proteins are anchored to the cell surface. The manuscript is well written and cover all the related aspect.

Author Response

Thanks for reviewing our manuscript.

Reviewer 2 Report

The manuscript is well organized and prepared. I only have a few technical comments / suggestions listed below.

  1. The Figures 1B and 4A would be more clear for readers if there were shown parts of agar plates with single colonies.
  2. In the Figures 2, 3 and 5D, there are ladders with values but without units (Da, kDa?).
  3. In the line 335, at the end of sentence "Previously, Vincent et al, showed that PorP interacts with PorK using coimmunoprecipitations in E. coli", there should be given reference number (27?).
  4. Line 440 includes "complete protease inhibitors". Provide more information on these inhibitors (e.g. dealer name).
  5. Immunoblot analysis description should include information about: how many times were antibodies diluted ? how long was membrane incubated with antibodies and what was the temperature of incubation? 
  6. In the lines 507-509, there should be given a more detailed description of conditions of  incubation with anti-
    PG1035 as well as with the secondary antibodies (antibodies dilution, time of incubation, temperature)

Author Response

The manuscript is well organized and prepared. I only have a few technical comments / suggestions listed below.

  1. The Figures 1B and 4A would be more clear for readers if there were shown parts of agar plates with single colonies.

In our experience, showing the pigmentation of single colonies is clearer if there is doubt in the acquisition of the pigmentation or if delayed pigmentation is observed in the mutant. We did not experience any of this as full pigmentation was observed within 7 days….Ln 119 and 226 

  1. In the Figures 2, 3 and 5D, there are ladders with values but without units (Da, kDa?).

This has been rectified in the figures and kDa has been added above the ladder.

  1. In the line 335, at the end of sentence "Previously, Vincent et al, showed that PorP interacts with PorK using coimmunoprecipitations in E. coli", there should be given reference number (27?).

 Apologies for this over-sight. The reference has now been added. Ln 346

  1. Line 440 includes "complete protease inhibitors". Provide more information on these inhibitors (e.g. dealer name).

The dealer name (Roche) has now been included. Ln 453

  1. Immunoblot analysis description should include information about: how many times were antibodies diluted ? how long was membrane incubated with antibodies and what was the temperature of incubation? 

These conditions have now been included from line 462-465

transferred onto a nitrocellulose membrane. The membrane was blocked in 5% skim milk for 1 h and probed with, mAb-1B5 (1:200) (a kind gift from Prof. M.A Curtis) [42], PorE [19] (1:3000), PG1035 (1:1000) or streptactin (1:2000) specific antibodies for 1 h at room temperature followed by anti-mouse or anti-rabbit HRP conjugated secondary antibodies (1:3000) for 1 h.

  1. In the lines 507-509, there should be given a more detailed description of conditions of  incubation with anti-
    PG1035 as well as with the secondary antibodies (antibodies dilution, time of incubation, temperature)

These conditions have now been included from Ln 522-523

, incubated with anti-PG1035 (1:200) for 40 min at 4oC, washed twice in Pg-PBS, incubated with Alexa Fluor 647-conjugated goat anti-rabbit mouse IgG (1:200) for 40 min at 4oC (Thermo Fisher Scientific) and washed again.

Reviewer 3 Report

Please refer the attached file for my comments

Author Response

The manuscript from Gorasia et al., covers the not so explored components of T9SS and provides important implications towards the type B CTD proteins anchorage to the cell surface.  Further, the authors cross linking results demonstrate the preliminary indications that PorE-PorP-PG1035 formation takes place independent of PorU and PorV.  The results are corroborated by similar finding in F. johnsoniae. I found the study interesting particularly due to the fact that type B CTD proteins has not been frequently explored and this manuscript provides important implications towards the type B CTD proteins anchorage mechanism. However, some points are not clear that authors must address. I have following comments to improve the manuscript:

  • It is not clear how many times the experiments were performed. If experiments were conducted in replicates or just a single time analysis was performed? No statistical analysis information is provided. Statistical significance of these results is not clear. Are these results reproducible?

All samples/experiments have been performed multiple times and the results are reproducible. Furthermore, the highlight of the work which is the complex formation between PorP-PorE-PG1035 have been shown in multiple ways: (i) Pull down with Streptactin in the PorEStrep strain, (ii) pull down with PG1035 antibodies in the WT strain and  (iii) the complex has also been shown by Blue Native PAGE.

  • Instead of writing the volume for loaded samples on the gel, mention the concentrations/amount.

The volumes have been changed to the protein amount in the full blot images

  • It is not clear what authors mean when they say the mutants (Eg line 391, porU, porV and porP), I had problem in understanding results and claims because I was not sure what was control in their experiments. Does it mean they are referring to the deletion mutants? If yes, mentions in M&M section How they were created? and add the delta symbol that is scientific way of writing deletion mutants?

A delta symbol has now been added in front of all the mutants that were originally written in italics with a lower case “p”. References were included in the manuscript to refer to the papers on how these mutants were created. Ln 402-403

  • The figures are not clear and informative. The figure needs to be re-organized. For example, Fig. 2 is this the one figure or two separate images? Since top (clearer) and below portions (messy) look quite different.

These are two different blots probed with two different antibodies. This has been shown in two different outlines. All the figures have been improved by using the same font.  

  • Bring supplementary Figure 1 in the main manuscript. Fig. 1B WT and PorEStrep looks similar.

The values of supplementary figure 1 have been tabulated in the main manuscript. Yes, WT and PorEstrep pigmentation are the same and are meant to look the same as already described in lines 104-105.

  • Bring corresponding gel images in the supplementary and add informative labels/legends. Also cite the gel images in the text for direct comparison to the main images in manuscript.

The images in the main manuscript have only been cropped to make figures more concise…therefore bringing the whole blot images as supplementary files will not add any value to the manuscript.

  • Emphasize the novelty of this study and provide speculation in the discussion/conclusion.

We have now included “for the first time” in Ln 25 and Ln 91 to emphasize the novelty of this study. Speculation was provided in the discussion, for example…on lines 304-305 and 386-389, we speculate that all type B CTD proteins bind to PorP-like proteins and also PorP and SprF may interact directly with Sov/SprA to collect its respective substrates.

The statements seem repetitive at some places, check the repetition. For example, Line 291 – 296 are similar to abstract.

                         We have changed the words in the abstract (Ln 22-24) to reduce the repetition.

Round 2

Reviewer 3 Report

The authors have made necessary amendments in response to the comments raise by me in previous review. I endorse the manuscript for publication in its current form.